# Symmetric Space Learning for Combinatorial Generalization

**Jaehyoung Jeong, Hee-Jun Jung, Kangil Kim**[*]
Department of AI Convergence
Gwangju Institute of Science and Technology
{jaehyoung98, jungheejun93}@gm.gist.ac.kr,
kangil.kim.01@gmail.com

## Abstract

Combinatorial generalization (CG)—generalizing to unseen combinations of known semantic factors—remains a fundamental challenge in machine learning. While symmetry-based methods are promising, they learn from observed data and thus fail at what we term **symmetry generalization**: extending learned symmetries to novel data. We address this by proposing a novel framework that endows the latent space with the structure of a **symmetric space**. This class of manifolds provides a principled geometric foundation for extending learned symmetries. Our method operates in two steps: first, it imposes this structure by learning the underlying algebraic properties via the **Cartan decomposition** of a learnable Lie algebra. Second, it uses **geodesic symmetry** as a powerful self-supervisory signal to ensure this learned structure extrapolates from observed samples to unseen ones. A detailed analysis on a synthetic dataset validates our geometric claims, and experiments on standard CG benchmarks show our method significantly outperforms existing approaches.

## 1 Introduction

Generalizing a model to unobserved combinations of semantic factors is a critical challenge in achieving human-like generalization (Fodor & Pylyshyn, 1988). In representation learning, this problem is known as Combinatorial Generalization (CG), where the goal is to capture the semantic data structure within latent representations (Vankov & Bowers, 2019). Despite its importance, most existing approaches often fail to achieve this effectively (Schott et al., 2021).

Symmetry, which captures transformations that leave an object's identity invariant, is a cornerstone for addressing CG. For instance, Higgins et al. (2022) demonstrated that symmetry learning effectively captures structural information, while Hwang et al. (2023) showed its direct benefits for CG. Nonetheless, a critical limitation persists: these methods learn symmetries exclusively from observed data, hindering their ability to generalize these symmetries to unseen combinations.

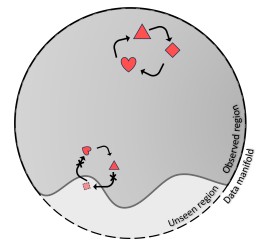

Figure 1: Since trained group actions should be closed on the observed region, they cannot affect the unseen region.

Exploiting geometric information offers a promising avenue for such generalization. Geometric Deep Learning has emerged as a powerful paradigm for understanding the dynamics of representation learning (Bengio et al., 2013; Bronstein et al., 2021). Specifically, the geometry of latent manifolds can provide deep insights into semantic factors and data relationships (Shao et al., 2018; Choi et al., 2021). However, integrating these geometric insights with symmetry learning has been limited, often relying on restrictive assumptions, such as predefined Lie groups or fully observed data, leaving a critical gap for CG.

---

[*]Corresponding author.

We summarize our primary contributions to the field of Combinatorial Generalization (CG) as follows:

1. **Problem Formulation:** We formally identify the core limitation of existing symmetry-based methods by defining the **Symmetry Generalization** challenge: extending learned symmetries from partially observed data to unobserved regions.

2. **Novel Framework:** We propose **CartanFM**, a novel framework that embeds the latent representation of transformations with the rigorous geometric structure of a **Symmetric Space**, offering a principled solution to the challenge.

3. **Technical Mechanisms:** We introduce two novel, complementary losses to realize this structure:

   - The **Cartan Loss** ($\mathcal{L}_{\text{Cartan}}$), which enforces the local algebraic structure via **Cartan Decomposition** on a learnable Lie algebra.
   - The **Geodesic Symmetry Consistency (GSC) Loss** ($\mathcal{L}_{\text{GSC}}$), a self-supervisory signal that leverages **geodesic symmetries** to explicitly extrapolate the learned structure to unseen regions of the data manifold.

## 2 BACKGROUND

A *manifold* is a topological space that is locally Euclidean. This concept is foundational to machine learning, motivated by the *manifold hypothesis*, which posits that high-dimensional data often lie on or near a low-dimensional manifold embedded within the ambient space (Narayanan & Mitter, 2010).

Many such manifolds exhibit rich symmetries that can be formally described using the theory of groups. A *Lie group* is a particularly suitable framework, as it is a differentiable manifold that is also a group with smooth operations. A canonical example is the general linear group $\text{GL}(n)$, the set of all invertible $n \times n$ matrices.

The interaction between a group and a manifold is formalized by the notion of a *homogeneous space*, which serves as a cornerstone of our method.

**Definition 2.1 (Homogeneous Space)** *A Riemannian manifold $\mathcal{M}$ is a **homogeneous space** if there exists a Lie group $G$ that acts transitively on $\mathcal{M}$.*

In a homogeneous space, any point can be mapped to any other point via an element of $G$, implying that the manifold is geometrically uniform. This property allows for a coordinate-free analysis of the manifold's structure through its symmetries (do Carmo, 1992).

While homogeneous spaces offer a general framework, our work focuses on a more structured subclass known as *symmetric spaces*. These spaces possess additional symmetries, such as point reflections, and can be defined from several equivalent perspectives (Helgason, 2001).

**Definition 2.2 (Symmetric Space via Geodesic Symmetry)** *Let $\mathcal{M}$ be a Riemannian manifold. The **geodesic symmetry** $s_p$ at a point $p \in \mathcal{M}$ is a map that reverses geodesics passing through $p$. Specifically, for a geodesic $\gamma(t)$ with $\gamma(0) = p$, it satisfies $s_p(\gamma(t)) = \gamma(-t)$. The manifold $\mathcal{M}$ is a **symmetric space** if, for every point $p \in \mathcal{M}$, this map $s_p$ can be extended to a global isometry.*

An equivalent algebraic characterization defines a symmetric space as a special type of coset space.

**Definition 2.3 (Symmetric Space via Lie Groups)** *Let $G$ be a connected Lie group and $K$ be a closed subgroup. The coset space $G/K$ is a **symmetric space** if there exists an involutive automorphism $\sigma : G \to G$ such that $G_0^\sigma \subseteq K \subseteq G^\sigma$, where $G^\sigma = \{g \in G | \sigma(g) = g\}$ and $G_0^\sigma$ is the identity component of $G^\sigma$.*

This group-level algebraic structure induces a corresponding structure at the Lie algebra level, giving rise to the third equivalent definition via the Cartan decomposition.

**Definition 2.4 (Symmetric Space via Cartan Decomposition)** *Let $\mathfrak{g}$ be the Lie algebra of $G$. A symmetric space structure on $G/K$ corresponds to a **Cartan decomposition** of its Lie algebra, $\mathfrak{g} = \mathfrak{k} \oplus \mathfrak{p}$, where $\mathfrak{k}$ is the Lie algebra of the subgroup $K$. This decomposition is induced by the*

*differential of $\sigma$ at the identity, an involutive automorphism $\theta$ on $\mathfrak{g}$. Here, $\mathfrak{k}$ and $\mathfrak{p}$ are the eigenspaces of $\theta$ for eigenvalues $+1$ and $-1$, respectively, and they satisfy the Lie bracket relations:*

$$[\mathfrak{k}, \mathfrak{k}] \subseteq \mathfrak{k}, \quad [\mathfrak{k}, \mathfrak{p}] \subseteq \mathfrak{p}, \quad and \quad [\mathfrak{p}, \mathfrak{p}] \subseteq \mathfrak{k}. \tag{1}$$

*The pair $(\mathfrak{g}, \mathfrak{k})$ is called a **symmetric pair**.*

## 3 METHOD

### 3.1 MOTIVATION: SYMMETRY GENERALIZATION

**Limitations of Existing Symmetry Learning**   Symmetry-based machine learning methods aim to capture data variations by learning underlying symmetries, often framed through equivariant group actions. While effective, these methods face a critical limitation when symmetries are learned exclusively from a subset of the data. For instance, approaches like Hwang et al. (2023) learn group actions from observed training samples to generate novel data. However, the learned symmetries are confined to the training data, restricting generalization to unseen samples. We term this challenge *symmetry generalization*, which we formalize as follows:

> **The Symmetry Generalization Challenge**
>
> Let $X$ be the complete data space and $X_{obs} \subset X$ be the observed subset. Let $G_{obs}$ be a symmetry group learned exclusively from actions within $X_{obs}$. The generalization failure occurs because for an unseen sample $x_{new} \in X \setminus X_{obs}$, there is generally no transformation $g \in G_{obs}$ and sample $x_{obs} \in X_{obs}$ such that $g \cdot x_{obs} = x_{new}$.

This limitation is not inherent in group theory itself, but in methods that infer symmetries from incomplete data. Overcoming this gap requires a framework capable of extending learned symmetries beyond the observed samples, motivating our use of geometric structures.

**Exploiting Geometric Information**   From a geometric perspective, the data manifold encodes the semantic structure of both observed and unobserved samples. A promising solution to the symmetry generalization challenge is to consider symmetries that act on the entire data manifold, not just on the training set. Imposing the structure of a *homogeneous space* provides a principled way to extend locally observed symmetries, as its transitive property guarantees a path from any observed point to any unobserved point. However, this is impractical for general problems because the global group actions required for such navigation are unknown.

To circumvent this, we propose enforcing the properties of a *symmetric space*—a special class of homogeneous spaces with additional geometric structure. A symmetric space is endowed with a **geodesic symmetry** (a reflection) at every point (Definition 2.2). This reflection provides a concrete, locally defined operation for generalization that does not require knowledge of the whole global group. Specifically, by reflecting an observed point through another chosen as a local origin, we can generate a plausible sample in an unseen region. By training our model to be consistent with this reflection property, we directly extend the learned symmetry structure beyond the training data.

### 3.2 SYMMETRIC SPACE LEARNING VIA LIE ALGEBRA

To model the data manifold $\mathcal{M}$ as a symmetric space $G/K$, we face two primary challenges. First, the algebraic structure, defined by the symmetry group $G$ and stabilizer $K$, is unknown for general datasets. Second, any structure learned solely from observed data must be able to generalize to unseen regions. Our method addresses these two challenges with two corresponding components for learning and generalizing the symmetric space structure.

**Learning the Symmetric Pair via Cartan Loss**   We begin by identifying the latent space of our feature encoder with the tangent space $T_o\mathcal{M}$ at a chosen origin $o$. In a symmetric space, this tangent space corresponds to the subspace $\mathfrak{p}$ from the Cartan decomposition $\mathfrak{g} = \mathfrak{k} \oplus \mathfrak{p}$ (Definition 2.4). For each data point $x$, the encoder thus outputs a latent vector $z_x \in \mathfrak{p}$. Crucially, our model also learns the bases for the subspaces $\mathfrak{k}$ and $\mathfrak{p}$ themselves, which are parameterized as sets of matrices. To

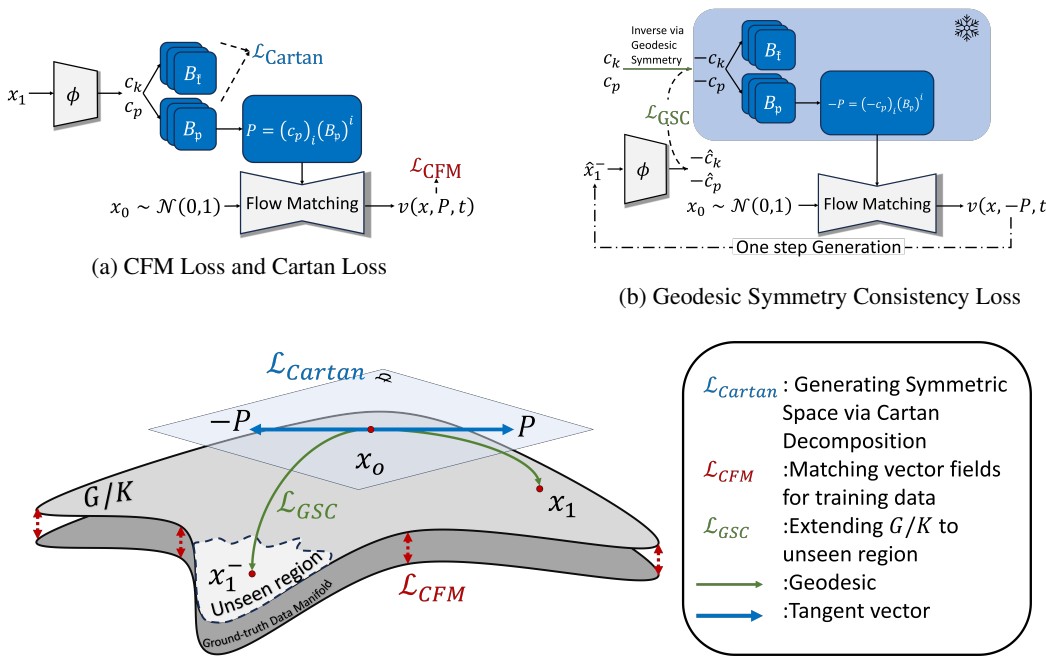

(a) CFM Loss and Cartan Loss

(b) Geodesic Symmetry Consistency Loss

(c) Geodesic Symmetry by Lie algebra negation.

Figure 2: Overview of our proposed CartanFM framework and its underlying geometric principle. **(a) Main Training Path:** The model learns to reconstruct data conditioned on a Lie algebra element $P$ with CFM Loss $\mathcal{L}_{\text{CFM}}$. The learnable bases that constitute $P$ are simultaneously regularized by the Cartan Loss ($\mathcal{L}_{\text{Cartan}}$) to form a valid algebraic structure. **(b) Generalization Path:** A self-supervisory loop leverages Geodesic Symmetry Consistency ($\mathcal{L}_{\text{GSC}}$) to train the encoder. This ensures that a synthetically generated unseen sample, when encoded, is consistent with the negated representation of its observed counterpart. **(c) Geometric Intuition:** In the learned symmetric space, geodesic symmetry corresponds to simple negation ($P \to -P$) in the tangent space, providing a principled bridge between the observed and unseen data regions.

ensure these learnable bases form a valid symmetric pair $(\mathfrak{g}, \mathfrak{k})$, we introduce the **Cartan Loss**, which imposes the required Lie bracket relations:

$$\mathcal{L}_{\text{Cartan}} := \mathcal{L}([\mathfrak{k}, \mathfrak{k}], \mathfrak{k}) + \mathcal{L}([\mathfrak{k}, \mathfrak{p}], \mathfrak{p}) + \mathcal{L}([\mathfrak{p}, \mathfrak{p}], \mathfrak{k}), \qquad (2)$$

where $\mathcal{L}(\cdot, \cdot)$ measures the projection error of the Lie bracket's output onto the target subspace.

**Generalizing Learned Structures via Geodesic Symmetry Consistency** The algebraic structure learned via the Cartan Loss only reflects the observed data. To promote generalization, we leverage a key property of symmetric spaces: the geodesic symmetry $s_o$ at the origin corresponds to negation in the tangent space $\mathfrak{p}$. Specifically, if a data point $x_1$ is represented by the tangent vector $p \in \mathfrak{p}$, its symmetric counterpart $x_2 = s_o(x_1)$ is defined by $-p \in \mathfrak{p}$. We operationalize this geometric prior through a cycle-consistency objective we term *Geodesic Symmetry Consistency* (GSC), with the following procedure:

1. Encode an observed sample $x_{obs}$ into its latent vector: $p_{obs} = \text{Encoder}(x_{obs}) \in \mathfrak{p}$.

2. Generate an unseen "candidate" sample by decoding the negated vector: $x_{cand} = \text{Decoder}(-p_{obs})$.

3. Encode the candidate sample back into the latent space: $\hat{p}_{cand} = \text{Encoder}(x_{cand})$.

4. Optimize the GSC Loss to enforce consistency between the re-encoded and the negated original vectors:

$$\mathcal{L}_{\text{GSC}} := \mathbb{E}_{x_{obs} \sim X_{obs}} \left[ \|\hat{p}_{cand} + p_{obs}\|^2 \right]. \qquad (3)$$

This objective encourages the model to map geometrically symmetric points in the data space to algebraically opposite points in the latent space, thereby extending the learned structure to unseen regions.

### 3.3 IMPLEMENTATION DETAILS

**Overview**   This section outlines the practical implementation of our symmetric space learning framework. We integrate our proposed architectural components and loss functions into a state-of-the-art generative model by conditioning its underlying vector field. As illustrated in Figure 2, our model consists of three core components:

- **A Lie Algebra Encoder ($\phi$):** Maps an input data point to the coefficients $(c_{\mathfrak{k}}, c_{\mathfrak{p}})$ for the learnable Lie algebra bases.
- **Learnable Lie Algebra Bases ($B_{\mathfrak{k}}, B_{\mathfrak{p}}$):** A set of matrices that form the bases for the subspaces $\mathfrak{k}$ and $\mathfrak{p}$, respectively. The Cartan Loss regularizes these.
- **A Conditional Flow Matching Model:** Generates a vector field $v(x, t, P)$ that is conditioned on the tangent vector $P \in \mathfrak{p}$, which is constructed from the encoder's output.

These components are trained jointly. The generative capability is learned via the standard Conditional Flow Matching loss ($\mathcal{L}_{\mathrm{CFM}}$). At the same time, the geometric structure is enforced by our two proposed losses, the Cartan Loss (Eq. 2) and the GSC Loss (Eq. 3). The following sections provide detailed explanations of each component.

**Backbone Generative Model**   We build our method upon Flow Matching (Lipman et al., 2022), a recent class of generative models that learn a probability flow from a noise distribution to the data distribution. This flow is defined by a neurally parameterized vector field $v(x, t, c)$ and is sampled by solving an ordinary differential equation (ODE). Inspired by DiffAE (Preechakul et al., 2022), which adapts diffusion models for representation learning, we employ an autoencoder architecture. A dedicated encoder network learns a mapping from a data sample $x$ to a Lie algebra element $P \in \mathfrak{p}$, which serves as the condition $c$ for the vector field, i.e., $v(x, t, P)$.

**Cartan Loss**   Our encoder consists of two main parts: a standard VAE-style encoder and a set of learnable basis matrices. For a given input, the encoder outputs coefficient vectors $c_{\mathfrak{k}} \in \mathbb{R}^{\mathrm{num}_{\mathfrak{k}}}$ and $c_{\mathfrak{p}} \in \mathbb{R}^{\mathrm{num}_{\mathfrak{p}}}$. These coefficients form linear combinations of the learnable bases $\{K_i\}$ and $\{P_j\}$ to produce Lie algebra elements $K = \sum_i c_{\mathfrak{k},i} K_i \in \mathfrak{k}$ and $P = \sum_j c_{\mathfrak{p},j} P_j \in \mathfrak{p}$. While $P$ directly conditions the generative process, the bases for both subspaces must satisfy the Lie bracket relations from Eq. 2. For computational efficiency, we apply these constraints directly to the basis elements. This is enforced via the Cartan Loss, which encourages orthogonality between the relevant subspaces:

$$\mathcal{L}_{\mathrm{Cartan}} = \sum_{i,j,l; i \neq j} |[K_i, K_j] \cdot P_l| + \sum_{i,j,l} |[K_i, P_j] \cdot K_l| + \sum_{i,j,l; i \neq j} |[P_i, P_j] \cdot P_l|. \qquad (4)$$

**Geodesic Symmetry Consistency**   To implement the GSC objective described in Eq. 3 efficiently, we bypass the computationally expensive complete decoding process of the Flow Matching model by using a one-step approximation. The core idea is to generate a synthetic sample corresponding to the negated latent vector $-P$ and ensure its re-encoded representation aligns with $-P$. We approximate this by taking a single step from the original data point $x_0$ along the vector field conditioned on $-P$. The resulting GSC Loss is:

$$\mathcal{L}_{\mathrm{GSC}} = \mathbb{E}\left[\|\mathrm{Encoder}(x_0 + (1 - \sigma_{\min}) \cdot v(x_0, t = 0, -P; \theta)) + P\|^2\right], \qquad (5)$$

where $\sigma_{\min}$ is a hyperparameter defined in Lipman et al. (2022). This objective pushes the encoder to be consistent with the geodesic symmetry defined in the latent space.

**Full Objective Function**   Our final training objective is a weighted sum of the generative loss and our architectural losses:

$$\mathcal{L} = \mathcal{L}_{\mathrm{CFM}} + \beta \cdot \mathcal{L}_{\mathrm{KL}} + \lambda_{\mathrm{Cartan}} \cdot \mathcal{L}_{\mathrm{Cartan}} + \lambda_{\mathrm{GSC}} \cdot \mathcal{L}_{\mathrm{GSC}} + \epsilon \cdot \mathcal{L}_{\mathrm{basis}} \qquad (6)$$

where:

- $\mathcal{L}_{\text{CFM}}$ is the conditional flow matching loss for the generative task (Lipman et al., 2022).

- $\mathcal{L}_{\text{KL}}$ is the standard Kullback–Leibler divergence from the VAE-style encoder.

- $\mathcal{L}_{\text{Cartan}}$ and $\mathcal{L}_{\text{GSC}}$ are our proposed losses for enforcing the symmetric space structure and encouraging generalization, respectively.

- $\mathcal{L}_{\text{basis}} = \sum_i 1/\|K_i\|_1 + \sum_j 1/\|P_j\|_1$ is a regularization term on the learnable basis elements $\{K_i\}$ and $\{P_j\}$ to prevent them from collapsing to zero.

- $\beta, \lambda_{\text{Cartan}}, \lambda_{\text{GSC}},$ and $\epsilon$ are hyperparameters that balance each term's contribution.

## 4  RELATED WORK

**Group-Theoretic Representation Learning**   Symmetry and group theory are cornerstones of modern representation learning (Higgins et al., 2022), particularly for achieving disentangled representations (Higgins et al., 2018). Various studies have demonstrated that enforcing group structures, such as orthogonality, enhances disentanglement (Cha & Thiyagalingam, 2023; Yang et al., 2022). This principle extends to Lie groups, which have been used to develop equivariant networks like G-CNNs on homogeneous spaces (Cohen et al., 2019) and their more general Lie algebra-based successors, L-CNNs (Dehmamy et al., 2021). Other works also leverage Lie groups to improve disentanglement (Zhu et al., 2021; Tonnaer et al., 2020) or to decouple object representations from group actions (Keurti et al., 2023). However, a common limitation of these "top-down" approaches is their reliance on predefined group structures, which restricts their ability to generalize from partially observed symmetries. In contrast, our framework learns the underlying algebraic structure of a symmetric space from data, enabling generalization to unobserved symmetry transformations.

**Geometry in Generative Models**   A geometric perspective offers powerful tools for improving generative models. Manifold learning, for instance, has been shown to enhance disentanglement in latent spaces (Fumero et al., 2021; Falorsi et al., 2018) and to model hierarchical data structures using hyperbolic geometry (Mathieu et al., 2019). Furthermore, geometric priors have proven effective for generalization tasks, such as improving out-of-distribution robustness (Ng et al., 2020; Vural & Guillemot, 2016). While these studies highlight the benefits of incorporating geometry, our work takes this a step further. We do not just learn an arbitrary manifold; we impose the rich structure of a *symmetric space*, which provides a principled framework for solving the *symmetry generalization* problem.

**Symmetry Discovery**   Finding the intrinsic symmetry structure is an important task across various domains, particularly in scientific data analysis. The Symmetry Discovery task aims to autonomously identify the latent structure of invariance and equivariance, which is explicitly unknown a priori. One efficient approach leverages Generative Adversarial Networks (GANs) to find Lie algebra bases that describe data distributions under various transformations (Yang et al., 2023b;a). Another promising approach utilizes infinitesimal generators, which are typically found using tools like Ordinary Differential Equations (ODEs) or self-supervised learning techniques (Shaw et al., 2024; Ko et al., 2024; Allingham et al., 2024; Hu et al., 2025). While these methods efficiently discover explicit symmetry structures using Lie algebras or vector fields, our approach focuses on how the model can extend learned symmetries rather than merely representing them. Moreover, given our crucial problem setting of partial observation (e.g., an incomplete manifold), there is no guarantee that such discovery methods can work properly without an explicit, global inductive bias. Thus, we propose our method to directly solve the *symmetry generalization* problem.

**Combinatorial Generalization**   Combinatorial generalization (CG) remains a critical challenge in machine learning (Vankov & Bowers, 2019). Early work suggested that disentangled representations could be a solution (Montero et al., 2020), but subsequent studies revealed their frequent failure to generalize to unseen combinations of factors (Montero et al., 2022). More recent approaches have focused on identifying sufficient conditions for CG (Wiedemer et al., 2023) or designing architectures that explicitly model group actions (Hwang et al., 2023). Our work contributes to this line of research by proposing a novel synthesis: we integrate latent geometry and group actions through the framework of symmetric spaces. This approach directly tackles the limitations of prior methods by addressing the fundamental problem of *symmetry generalization*.

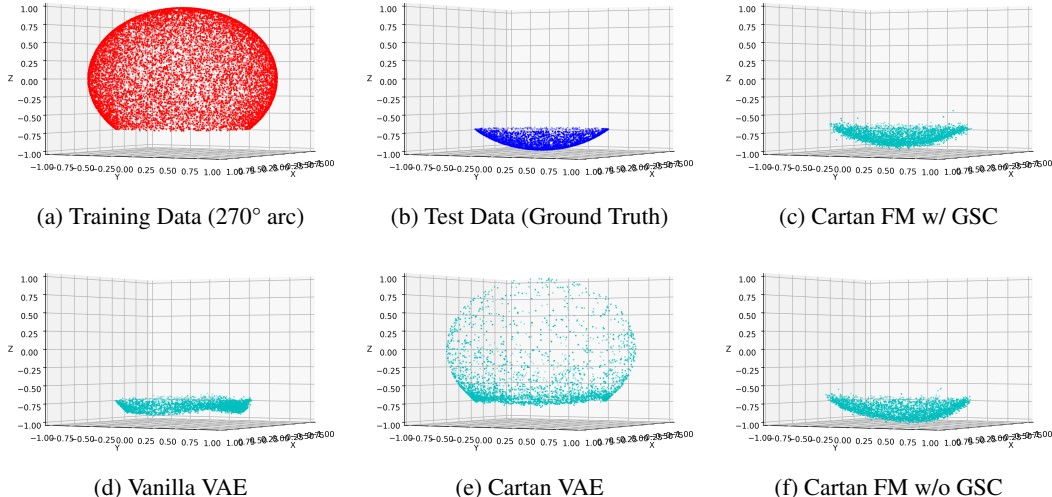

Figure 3: **Manifold Reconstruction Results on the sphere point cloud.** (a) The distribution of training data, consisting of a partial 270° arc. (b) The ground truth distribution for the unseen 90° arc test data. (c-f) Reconstruction results for the unseen test data from each model. Our full method (c) is the only one that successfully generalizes to the unseen region and reconstructs the complete sphere.

## 5 EXPERIMENTS

### 5.1 COMPREHENSIVE ANALYSIS VIA 3D SPHERE SHAPE MANIFOLD RECONSTRUCTION

**Experimental Setup**  As visualized in Figure 3, we partition a sphere point cloud to create a challenging generalization task. The **training set** consists of an incomplete 270-degree arc (Fig. 3a), while the **test set** is the entirely unobserved 90-degree arc (Fig. 3b). We compare four models based on the standard PointNet architecture (Qi et al., 2017):

- **Vanilla VAE:** A standard VAE baseline.

- **VAE + Cartan module:** A VAE equipped with our Cartan Loss, but without the generative flow model.

- **Flow Matching + Cartan Module:** Our full generative model, but without the GSC Loss, serving as a key ablation baseline.

- **Flow Matching + Cartan Module + GSC Loss:** Our full proposed model.

Table 1: Sphere Manifold Reconstruction Error (Chamfer distance)

| Method | Error ($\downarrow$) |
|---|---|
| Vanilla VAE | 0.0601 |
| Cartan + VAE | 0.5763 |
| Cartan + FM | 0.0068 |
| Cartan + FM + GSC | **0.0061** |

We evaluate reconstruction quality using the Chamfer distance (Fan et al., 2016), a standard metric for comparing point clouds. Full implementation details are available in Appendix C.1.

**Results**  The quantitative results in Table 1 show a clear progression across the model configurations. The **Vanilla VAE** baseline struggles to reconstruct the unseen test data, only managing to generate points near the observed boundary. The failure of the **VAE + Cartan module** underscores that the algebraic structure requires a compatible generative process. This result validates the design choice of using Flow Matching, as it operates on the tangent vector level and is thus ideally suited to realize the transformations defined by our learned Lie algebra. Accordingly, the **Flow Matching + Cartan Module** model achieves a significant performance improvement. Finally, adding the GSC Loss to create our full model, the **Flow Matching + Cartan Module + GSC Loss**, yields the best reconstruction performance by a large margin. This quantitative superiority is mirrored in the qualitative results (Figure 3), which visually demonstrate that our full model is uniquely capable of reconstructing the complete spherical manifold.

Table 2: Mean Squared Error($\downarrow$) in Combinatorial Generalization. Bold indicates the best performance.

| | Symmetry | Model | R2E | | | R2R | | |
|---|---|---|---|---|---|---|---|---|
| | | | Case1 | Case2 | Case3 | Case1 | Case2 | Case3 |
| dSprites | $\times$ | VAE | 7.25 | 8.08 | 14.96 | 28.81 | 34.64 | 28.90 |
| | | $\beta$-VAE ($\beta = 2$) | 12.29 | 13.91 | 24.23 | 31.02 | 35.74 | 60.31 |
| | | $\beta$-VAE ($\beta = 4$) | 22.54 | 20.13 | 31.18 | 32.38 | 40.80 | 61.43 |
| | | $\beta$-VAE ($\beta = 8$) | 31.02 | 27.83 | 39.65 | 39.76 | 55.30 | 67.03 |
| | | GAGA | 17.17 | 17.56 | 25.69 | 28.07 | 36.96 | 44.71 |
| | $\checkmark$ | CLGVAE | 72.40 | 76.49 | 133.15 | 132.83 | 218.15 | 236.28 |
| | | MAGANet | 13.30 | 10.63 | 19.49 | 115.46 | 108.96 | 23.35 |
| | | CartanFM (Ours) | **2.21** | **1.10** | **2.49** | **7.02** | **9.57** | **5.87** |
| 3D Shapes | $\times$ | VAE | 10.93 | 13.82 | 12.40 | 21.07 | 37.65 | 13.65 |
| | | $\beta$-VAE ($\beta = 2$) | 12.70 | 17.34 | 18.35 | 33.82 | 41.42 | 17.71 |
| | | $\beta$-VAE ($\beta = 4$) | 22.17 | 22.21 | 22.81 | 46.05 | 49.06 | 22.45 |
| | | $\beta$-VAE ($\beta = 8$) | 21.11 | 48.52 | 46.55 | 99.73 | 63.85 | 44.57 |
| | $\checkmark$ | CLGVAE | 15.54 | 71.13 | 86.60 | 75.73 | 38.63 | 15.46 |
| | | MAGANet | 16.79 | 16.66 | 19.60 | 23.69 | 37.79 | 18.87 |
| | | CartanFM (ours) | **5.20** | **6.96** | **5.99** | **8.12** | **33.87** | **6.68** |
| MPI3D | $\times$ | VAE | 7.80 | 12.69 | 6.51 | 6.29 | 6.91 | 9.35 |
| | | $\beta$-VAE ($\beta = 2$) | 20.03 | 72.03 | 7.49 | 15.68 | 20.40 | 13.02 |
| | | $\beta$-VAE ($\beta = 4$) | 30.95 | 73.41 | 8.35 | 17.05 | 20.61 | 13.36 |
| | | $\beta$-VAE ($\beta = 8$) | 33.85 | 73.95 | 10.02 | 18.35 | 22.87 | 15.22 |
| | $\checkmark$ | MAGANet | 10.97 | 21.56 | 6.56 | 8.12 | 7.71 | 8.42 |
| | | CartanFM (ours) | **1.18** | **1.99** | **0.51** | **0.51** | **0.76** | **0.50** |

## 5.2 PERFORMANCE ON COMBINATORIAL GENERALIZATION BENCHMARKS

**Datasets and Protocol** We evaluate our method on two standard benchmarks for combinatorial generalization, dSprites (Matthey et al., 2017) and 3D Shapes (Burgess & Kim, 2018), as well as a more complex benchmark, MPI3D (Gondal et al., 2019). Following the protocol of Montero et al. (2020), we create training and test splits for each dataset under two challenging settings: *Recombination-to-Elements* (R2E) and *Recombination-to-Range* (R2R). To ensure robust evaluation and minimize bias from any single data split, we generate and test on three distinct sets of excluded factor combinations for both R2E and R2R settings. Further details are in Appendix C.2.

**Baselines and Training Details** We compare our method against VAE-based baselines from two categories: (1) models that do not explicitly model symmetries, VAE (Kingma & Welling, 2013) and $\beta$-VAE (Higgins et al., 2016), Geometry Aware Generative Autoencoder (GAGA) (Sun et al., 2025); and (2) symmetry-aware models, Commutative Lie Group VAE (CLGVAE) (Zhu et al., 2021) and MAGANet (Hwang et al., 2023). All models were trained for 100 epochs using an Adam optimizer with a learning rate of 5e-4. For $\beta$-VAE, we report the best performance for each $\beta \in \{2, 4, 8\}$. For MAGANet, we conducted a grid search to determine the optimal hyperparameters, as we were unable to reproduce the reported performance from the original paper. To ensure a fair comparison, all models were trained and evaluated using the Mean Squared Error (MSE) loss.

**Quantitative Analysis** Table 2 presents the quantitative MSE results for all tasks. Our method, CartanFM, demonstrates superior performance across all three datasets. **On dSprites**, CartanFM consistently outperforms all baselines in both R2E and R2R settings. The improvement is particularly pronounced in the more challenging R2R setting. **On 3D Shapes**, which contains more complex factors, CartanFM again establishes a clear and significant advantage, outperforming all baselines across every case. **On MPI3D**, the most complex benchmark, CartanFM achieves state-of-the-art results by a large margin in all settings. This substantial performance gain on a larger, more varied dataset suggests that our method effectively leverages more data to learn the underlying symmetric space structure better.

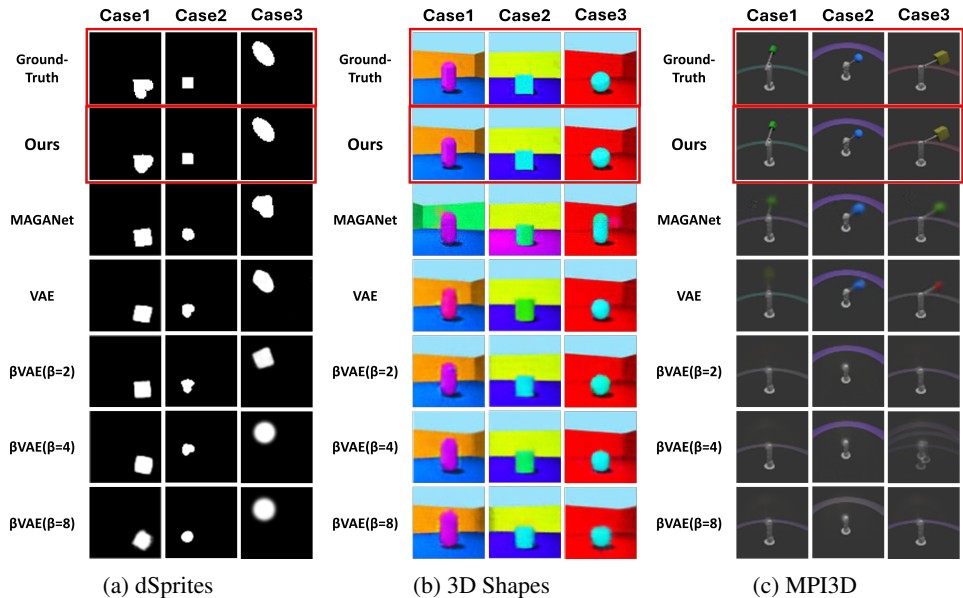

Figure 4: Generated Images in Recombination-to-Range (R2R) Setting

Table 3: Ablation study of our core components. Bold indicates the best performance. In the 'Components' columns, ✓ denotes the inclusion and × denotes the exclusion of the corresponding loss term.

| | Components | | R2E | | | R2R | | |
|---|---|---|---|---|---|---|---|---|
| | $\mathcal{L}_{\text{Cartan}}$ | $\mathcal{L}_{\text{GSC}}$ | Case1 | Case2 | Case3 | Case1 | Case2 | Case3 |
| dSprites | × | × | 1.87 | 1.89 | 4.52 | 29.06 | 23.70 | 10.40 |
| | ✓ | × | 1.85 | 1.64 | 21.50 | 57.38 | 18.52 | 40.34 |
| | × | ✓ | **1.31** | **1.09** | 2.94 | 13.11 | 47.43 | 9.23 |
| | ✓ | ✓ | 2.21 | 1.10 | **2.49** | **7.02** | **9.57** | **5.87** |
| 3D Shapes | × | × | 9.70 | 9.25 | 9.20 | 20.12 | 33.09 | 8.97 |
| | ✓ | × | 6.00 | 9.32 | 12.50 | 9.42 | **32.61** | 10.96 |
| | × | ✓ | 5.96 | 8.30 | 8.85 | 8.96 | 33.16 | 12.11 |
| | ✓ | ✓ | **5.20** | **6.96** | **5.99** | **8.12** | 33.87 | **6.68** |

**Qualitative Analysis**  Figure 4a, Figure 4b, and Figure 4c show the generated images for the R2R setting across different datasets and models. For VAEs, the generated outputs often appear as blurry or distorted blobs, particularly at higher values of $\beta$. By contrast, our method effectively captures critical factors, such as distinct object shapes, where baseline models struggle to generalize to unseen cases. For example, as shown in Fig. 4a, our method successfully generates heart shapes, whereas other methods produce squares, ellipses, or amorphous blobs. Similarly, as illustrated in Fig. 4b, VAEs generate cylinders instead of cubes, indicating a failure to capture the underlying factors of variation. Furthermore, as shown in Fig. 4c, our method captures the object's shape, color, and background more accurately than other methods. Our approach demonstrates strong generalization capabilities, effectively generating unseen combinations while preserving geometric and semantic fidelity.

**Ablation Study**  We conduct an ablation study to analyze the individual contributions of our two main components: the **Cartan Loss** ($\mathcal{L}_{\text{Cartan}}$) for inducing the symmetric space structure, and the **Geodesic Symmetry Consistency (GSC) Loss** for extending it. We evaluate four model configurations: the Flow Matching backbone alone, the backbone with each of our two losses applied individually, and our full model combining both.

As summarized in Table 3, the backbone model without our components struggles in most cases. Adding either the Cartan Loss or the GSC Loss individually yields significant performance gains, demonstrating that both are effective for improving generalization. The GSC Loss alone provides a particularly substantial boost, highlighting the importance of extending the learned structure. Our full model, which combines both losses, achieves the best or highly competitive performance in nearly all scenarios. This result confirms that the two components are complementary and that their synergy is crucial for the robust performance of CartanFM.

This study addressed the limitations of trained symmetries in generalizing to unseen data for combinatorial generalization. We proposed a novel method for learning a symmetric space structure on the data manifold and extending it to unseen data via Lie algebra properties and geodesic symmetry consistency, which facilitates the generalization of trained symmetries. An in-depth analysis of a synthetic sphere manifold dataset validates our geometric claims and the effectiveness of our approach. Furthermore, experiments on widely used benchmarks, including dSprites, 3D Shapes, and MPI3D, corroborated that our method significantly outperforms existing approaches. Our study is the first to establish the utility of integrating manifold and symmetry learning to enhance combinatorial generalization. This contribution opens up promising directions for future research, including the exploration of diverse sampling strategies tailored to specific data characteristics and the extension of the approach to a broader range of generalization tasks beyond combinatorial generalization.

## 6 CONCLUSION

This study addressed the limitations of trained symmetries in generalizing to unseen data for combinatorial generalization. We proposed a novel method for learning a symmetric space structure on the data manifold and extending it to unseen data via Lie algebra properties and geodesic symmetry consistency, which facilitates the generalization of trained symmetries. An in-depth analysis of a synthetic sphere manifold dataset validates our geometric claims and the effectiveness of our approach. Furthermore, experiments on widely used benchmarks, including dSprites, 3D Shapes, and MPI3D, corroborated that our method significantly outperforms existing approaches. Our study is the first to establish the utility of integrating manifold and symmetry learning to enhance combinatorial generalization. This contribution opens up promising directions for future research, including the exploration of diverse sampling strategies tailored to specific data characteristics and the extension of the approach to a broader range of generalization tasks beyond combinatorial generalization.

### ACKNOWLEDGMENTS

This research was supported by Culture, Sports and Tourism R&D Program through the Korea Creative Content Agency grant funded by the Ministry of Culture, Sports and Tourism in 2022 (Project Name: Development of service robot and contents supporting children's reading activities based on artificial intelligence, Project Number: R2022060001, Contribution Rate: 33%). This work was supported by Institute of Information & communications Technology Planning & Evaluation (IITP) grant funded by the Korea government (MSIT) (No. 2019-0-01842, Artificial Intelligence Graduate School Program (GIST), Contribution Rate: 10%). This work was supported by the National Research Foundation of Korea (NRF) grant funded by the Korea government (MSIT) (No. 2022R1A2C2012054, Development of AI for Canonicalized Expression of Trained Hypotheses by Resolving Ambiguity in Various Relation Levels of Representation Learning, Contribution Rate: 56%).

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

## A  Detail on Theoretical Background

**Group Action**    Group $(G, *)$ is a mathematical structure which is a tuple of a set $G$ and a binary operation $*$ closed on the set. The group should satisfy the following axioms:

1. **(associativity)** $a * (b * c) = (a * b) * c$
2. **(identity element)** there exists $e \in G$ such that $a * e = e * a$
3. **(inverse element)** there exists $a^{-1} \in G$ such that $a * a^{-1} = a^{-1} * a = e$

for every $a, b, c \in G$. The group plays as a representation of symmetry. The group action on a set $X$ of a group $G$ is a map $f : G \times X \to X$ which satisfies the following axioms:

1. **(identity)** $f(e, x) = x$
2. **(compatibility)** $f(g, f(h, x)) = f(gh, x)$

for an identity $e \in G$ and every $g, h \in G$ and every $x \in X$. We can decompose natural phenomena into objects and symmetries of those via group action.

**Definition A.1** *Let $G$ be a group and $X$ be a $G$-space. The action is said to be transitive if there exists $g \in G$ such that $g * x = y$ for any $x, y \in X$.*

This means that every point $x \in X$ can be translated into any point in $X$ with an action $g \in G$.

**Geometry**    In geometry and topology, a manifold is a topological space that locally resembles Euclidean space at every point. More formally, a smooth manifold, which is a type of manifold, can be defined as follows (do Carmo, 1992).

**Definition A.2** *A smooth (or differentiable) manifold of dimension $n$ is a set $\mathcal{M}$ and a family of injective mappings $x_\alpha : U_\alpha \subset \mathbb{R}^n \to \mathcal{M}$ of open sets $U_\alpha$ of $\mathbb{R}^n$ into $\mathcal{M}$ such that*

1. $\bigcup_\alpha x_\alpha(U_\alpha) = \mathcal{M}$.

2. *for any pair $\alpha, \beta$ with $x_\alpha(U_\alpha) \cap x_\beta(U_\beta) = W \neq \emptyset$, the sets $x_\alpha^{-1}(W)$ and $x_\beta^{-1}(W)$ are open sets in $\mathbb{R}^n$ and the mappings $x_\beta^{-1} \circ x_\alpha$ are differentiable.*

3. *The family $\{(U_\alpha, x_\alpha)\}$ is maximal relative to the conditions (1) and (2).*

Every point on a manifold has a tangent space, which is the vector space tangent to the manifold.

**Definition A.3** *Let $\mathcal{M}$ be a differentiable manifold. A differentiable function $\alpha : (-\epsilon, \epsilon) \to \mathcal{M}$ is called a (differentiable) curve in $\mathcal{M}$. Suppose that $\alpha(0) = p \in \mathcal{M}$, and let $D$ be the set of functions on $\mathcal{M}$ that are differentiable at $p$. The tangent vector to the curve $\alpha$ at $t = 0$ is a function $\alpha' : D \to \mathbb{R}$ given by*

$$\alpha'(0)f = \left.\frac{d(f \circ \alpha)}{dt}\right|_{t=0}, f \in D. \tag{7}$$

*A tangent vector at $p$ is the tangent vector at $t = 0$ of some curve $\alpha : (-\epsilon, \epsilon) \to \mathcal{M}$ with $\alpha(0) = p$. The set of all tangent vectors to $\mathcal{M}$ at $p$ will be indicated by the tangent space $T_p\mathcal{M}$.*

Informally, a *neighborhood* of a point $p$ on a manifold is an open subset of $\mathcal{M}$ that contains $p$.

## B  Architectures

### B.1  Baselines

To implement $\beta$-VAE (Higgins et al., 2016), we used the structure introduced in (Burgess et al., 2018). The encoder consists of four convolutional layers with 32 channels, two fully connected layers with 256 nodes, and a fully connected layer with $d$ nodes, where $d$ is the latent vector dimension. The

decoder consists of a transpose of the encoder structure. ReLU activation is used for each layer, except for the last layer of the encoder and decoder.

To implement MAGANet (Hwang et al., 2023), we follow the proposed architecture. The encoder for modeling the group actions was the same as that of the VAE encoder architecture. The decoder consists of a linear layer without bias to apply the group action, as well as the GLOW model (Kingma & Dhariwal, 2018). The GLOW model comprises three flow modules, each consisting of three flow blocks and a squeeze layer. Each flow block comprised ActNorm, $1 \times 1$ convolution without LU decomposition, and an additive coupling layer. MAGANet incorporates three primary loss functions to train the VAE and flow-based components.

$$
\begin{align}
\mathcal{L}_{\text{recon}} &= l_{\mathcal{D}}(D(E(x_1, x_2), x_1), x_2), \tag{8} \\
\mathcal{L}_{\text{recon\_latent}} &= l_1(E(x, D(z, x)), z), \tag{9} \\
\mathcal{L}_{\text{Base}} &= \mathcal{L}_{\text{recon}} + \beta_{\text{KL}}\mathcal{L}_{\text{KL}} + \beta_{\text{recon\_latent}}\mathcal{L}_{\text{recon\_latent}}, \tag{10}
\end{align}
$$

where $l_{\mathcal{D}}$ denotes the loss in image space, $l_1$ represents the $L_1$ norm, $E$ is the encoder, and $D$ is the decoder. The hyperparameters $\beta_{\text{KL}}$ and $\beta_{\text{recon\_latent}}$ control the weighting of KL divergence and latent reconstruction losses, respectively.

### B.2 Proposed Method

We utilize the Lie algebra encoder, inspired by Zhu et al. (2021). However, we do not employ commutativity, as most symmetric spaces are not commutative, and the influence of commutativity on combinatorial generalization is not our research focus. The flow matching module consists of a simple UNet architecture and conditioning by the Lie algebra using AdaGN, as proposed in Dhariwal & Nichol (2021).

## C Experiments Details

### C.1 Experiment on Sphere Synthetic Data

**Dataset and Common Setting**  The data point cloud comprises 15,000 points of the training data and 3,000 points of the test data. Every model was trained for 100 epochs with a learning rate of 1e-4. Each model has three latent dimensions; Cartan models have two $B_{\mathfrak{p}}$ and a $B_{\mathfrak{k}}$.

**Model Architecture**  Every PointNet block consists of layers of point-wise MLPs and Leaky ReLUs between the layers. Moreover, for flow matching, the AdaGN layer conditioning is performed with Lie algebra after each block.

**Hyperparameter**  $\beta$ for the VAE encoder is set to 0.001 to prevent the model from collapsing. $\lambda_{\text{Cartan}}$ is set to 0.1 and $\epsilon$ in Equation 6 is set to 0.001. The number of groups in AdaGN is set to 8.

### C.2 Experiment on Benchmarks of Combinatorial Generalization

**Dataset Setting**  For split dSprites dataset (Matthey et al., 2017) in Recombination-to-Elements setting, we except following combinations:

1. shape=ellipse, scale=0.5, $120° \leq$ orientation $\leq 240°$, $0.6 < x$, $0.6 < y$,
2. scale=0.5, orientation=$0°$, $x \leq 0.25$, $y \leq 0.25$,
3. shape=heart, orientation=$0°$, $0.5 < x$, $0.5 < y$.

In the Recombination-to-Range setting, we except the following combinations:

1. shape=heart, $0.5 < x$,
2. shape=square, $0.5 < x$,
3. shape=ellipse, $3 <$scale, $y < 0.5$.

For 3D Shapes dataset Burgess & Kim (2018) in Recombination-to-Elements setting, we except following combinations:

1. floor-hue$> 0.5$, wall-hue$> 0.5$, object-hue$> 0.5$, scale=7, shape=cube, orientation=$0°$,

2. floor-hue$\leq 0.5$, wall-hue$\leq 0.5$, object-hue$\leq 0.5$, scale=7, shape=cylinder, orientation=$0°$,

3. floor-hue$\leq 0.5$, wall-hue$> 0.5$, object-hue=0, scale=0, shape=[sphere, cube], orientation=$-30°$.

In the Recombination-to-Range setting, we except the following combinations:

1. $0 \leq$ floor-hue$\leq 1$, $0 \leq$ wall-hue$\leq 1$, object-hue$> 0.5$, $0 \leq$ scale$\leq 1$, shape=oblong, $-30° \leq$ orientation$\leq 30°$,

2. $0 \leq$ floor-hue$\leq 1$, $0 \leq$ wall-hue$\leq 1$, $0 \leq$ object-hue$\leq 1$, scale$\leq 2$, shape=sphere, $-30° \leq$ orientation$\leq 30°$,

3. floor-hue$< 0.5$, $0 \leq$ wall-hue$\leq 1$, $0 \leq$ object-hue$\leq 1$, $0 \leq$ scale$\leq 8$, shape=cylinder, $-30° \leq$ orientation$\leq 0°$.

## C.3    Model Architectures

### C.3.1    Baseline Models

$\beta$-**VAE**    For the $\beta$-VAE baseline (Higgins et al., 2016), we adopt the standard convolutional architecture from Burgess et al. (2018).

- **Encoder:** Consists of four convolutional layers (32 channels, kernel size 4, stride 2), followed by two fully-connected layers (256 nodes each) that output a latent vector of dimension $d$.
- **Decoder:** Symmetrically mirrors the encoder architecture using transposed convolutional layers.
- **Activation:** ReLU activation is used for all layers except for the output layers of the encoder and the decoder.

**MAGANet**    For MAGANet (Hwang et al., 2023), the encoder for learning the group action is identical to the $\beta$-VAE encoder. The decoder is composed of a linear layer (without bias) to apply the group action and a GLOW model (Kingma & Dhariwal, 2018) for the generative process. The GLOW model comprises three flow modules, each consisting of three blocks that utilize ActNorm, a $1 \times 1$ convolution, and an additive coupling layer. The model is trained with the following primary loss functions:

$$\mathcal{L}_{\text{recon}} = l_{\mathcal{D}}(D(E(x_1, x_2), x_1), x_2), \tag{11}$$

$$\mathcal{L}_{\text{recon\_latent}} = l_1(E(x, D(z, x)), z), \tag{12}$$

$$\mathcal{L}_{\text{Base}} = \mathcal{L}_{\text{recon}} + \beta_{\text{KL}}\mathcal{L}_{\text{KL}} + \beta_{\text{recon\_latent}}\mathcal{L}_{\text{recon\_latent}}, \tag{13}$$

where $l_{\mathcal{D}}$ is the image reconstruction loss, and $l_1$ is the $L_1$ norm.

### C.3.2    Proposed Method (CartanFM)

Our model's architecture consists of a Lie algebra encoder (detailed in Section 3.3) and a conditional generative model.

**Generative Model**    The generative model is a simple UNet architecture, where the conditioning on the Lie algebra element $P$ is performed using Adaptive Group Normalization (AdaGN) (Dhariwal & Nichol, 2021) in each block. The channel dimensions are structured as follows:

- **Input Channels:** 1 (dSprites, MPI3D) or 3 (3D Shapes).
- **Downsampling Path Channels:** $[C, 64, 128, 256]$, where $C$ is the input channel size.
- **Middle Block Channels:** $[256]$.
- **Upsampling Path Channels (with skip connections):** $[256 + 256, 128 + 128, 64 + 64, C + C]$.

**ODE Solver**    For the decoding process, we use the standard Euler method, a basic ODE solver implemented in the `torchdiffeq` library (Chen, 2018).

**Hyperparameter**    Hyperparameters of MAGANet had been searched by grid and Bayesian search in the range of $\beta_{\mathrm{KL}}, \beta_{\mathrm{LR}} \in [0, 1000]$. We find that 1 is the best value for hyperparameters.

The proposed model has the following hyperparameter settings:

- $\lambda_{\mathrm{Cartan}}, \lambda_{\mathrm{GSC}} = 1.0$,
- $\epsilon = 0.001$,
- $\beta = 0.01$,
- Group number of AdaGN is 8.

We set the number of basis elements for $\mathfrak{p}$ to 10, consistent with the latent dimension of the baseline models, and the number of basis elements for $\mathfrak{k}$ to 5.

**Computing Resource**    We conducted experiments on a local server equipped with NVIDIA graphics cards, including the RTX 2080 Ti, RTX 3090, RTX A6000, and RTX A100. Each run requires approximately 6000MiB of VRAM and takes about 30 hours. These requirements may vary depending on the dataset, split settings, and GPU used.

### C.4    ADDITIONAL GENERATED SAMPLES

Every image consists of ground truth (odd columns) and reconstructions (even columns).

## D    ADDITIONAL EXPERIMENTS

### D.1    LIEGAN ON SPHERE DATASET

We conducted additional experiments of LieGAN (Yang et al., 2023a) on 3D Sphere dataset of subsection 5.1. Experiments details are as follows:

- Training dataset is composed of 15000 points on upper sphere of 135 degree.
- Lie algebra basis of model is randomly initialized.
- Discriminator consists of PointNet block and linear layers.

Sampled result image can be found in 10. Because of difference between VAE based model and GANs, we sample transformed points by sampled 100 transformations and inspect whether test region can be generated by such transformations. We can observe that test region is not generated perfectly by such transformation. Moreover, learned basis are not form skew-symmetric.

### D.2    VERIFYING GEODESIC SYMMETRY

We visualize pairs of basis vectors (lines ending in arrows) and their negated vectors (lines ending in circles). This alignment is consistent with the Lie algebra-theoretic definition of geodesic symmetry, implying that points on the generated geodesics are effectively reflected through the origin to their antipodal position including test region.

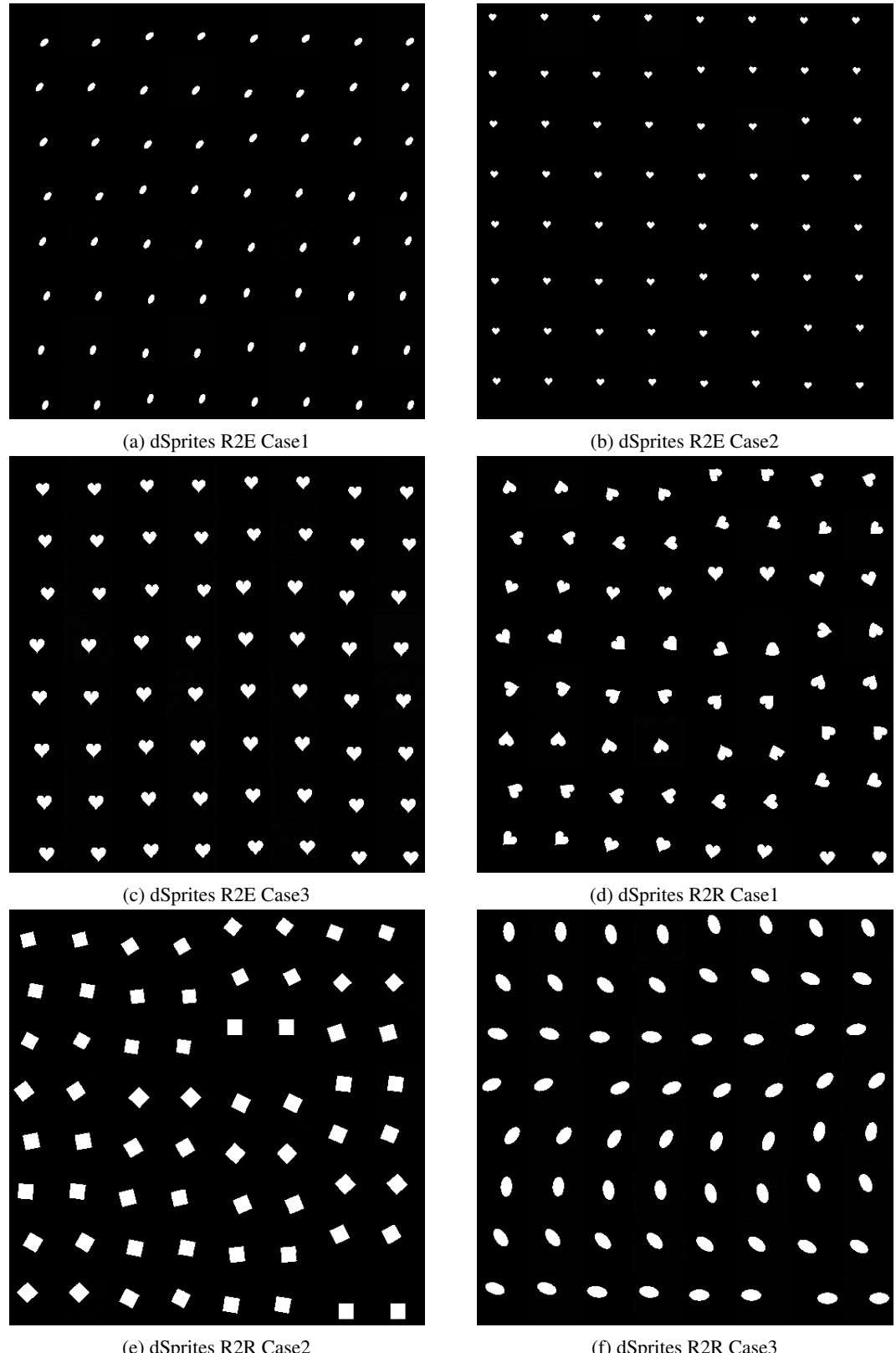

(a) dSprites R2E Case1

(b) dSprites R2E Case2

(c) dSprites R2E Case3

(d) dSprites R2R Case1

(e) dSprites R2R Case2

(f) dSprites R2R Case3

Figure 5: Generated Images dSprites

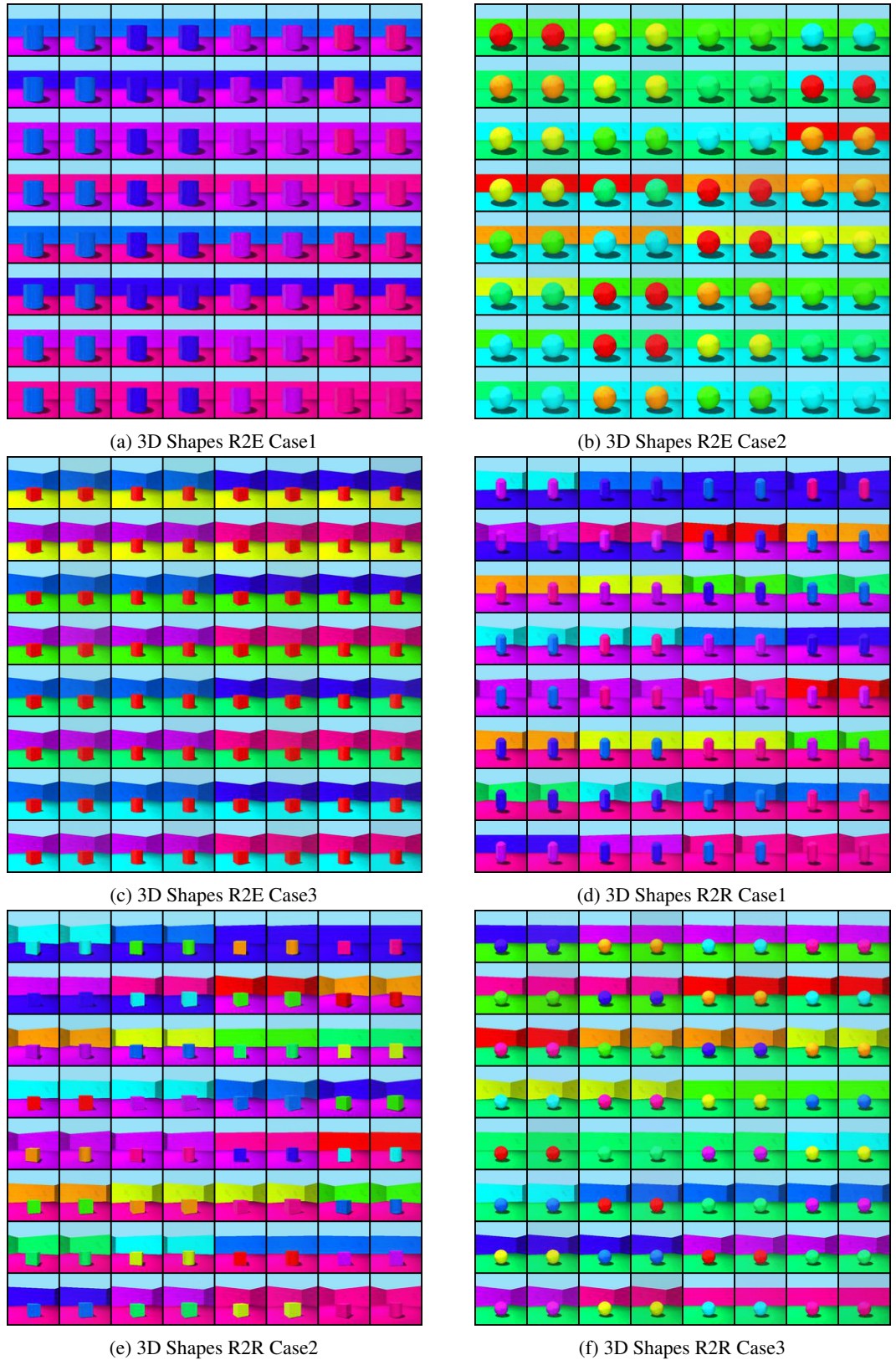

(a) 3D Shapes R2E Case1

(b) 3D Shapes R2E Case2

(c) 3D Shapes R2E Case3

(d) 3D Shapes R2R Case1

(e) 3D Shapes R2R Case2

(f) 3D Shapes R2R Case3

Figure 6: Generated Images 3D Shapes

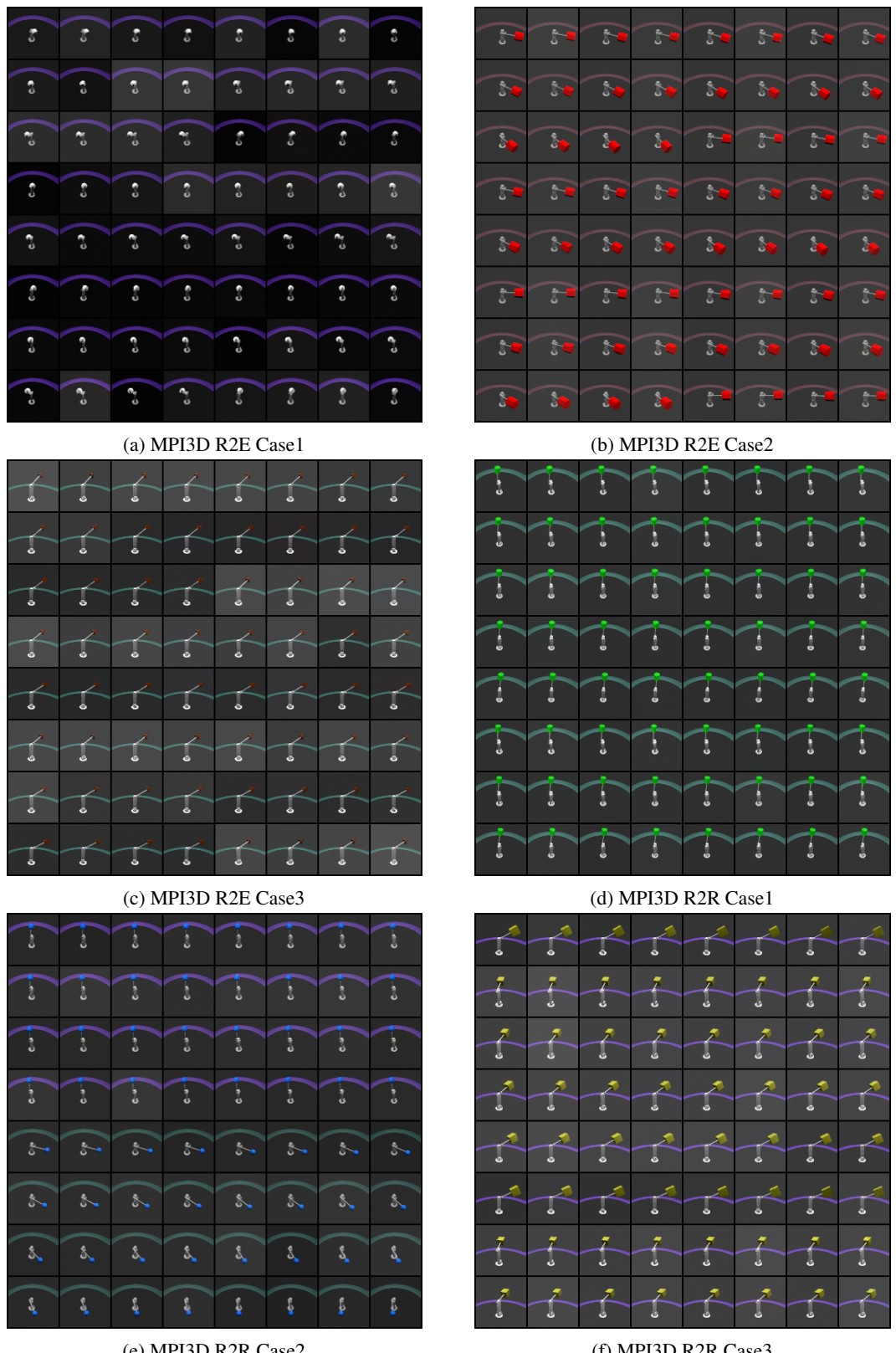

(a) MPI3D R2E Case1

(b) MPI3D R2E Case2

(c) MPI3D R2E Case3

(d) MPI3D R2R Case1

(e) MPI3D R2R Case2

(f) MPI3D R2R Case3

Figure 7: Generated Images MPI3D

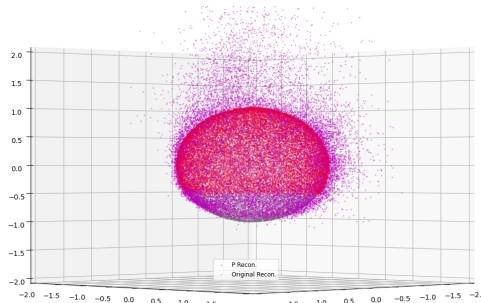

Figure 8: Visualization of Geodesic Symmetries

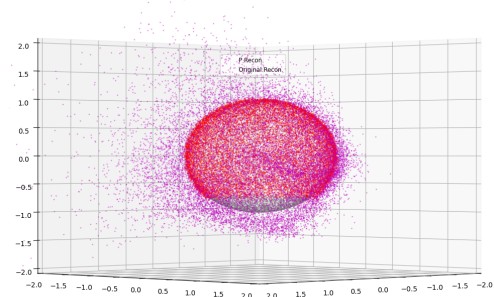

Figure 9: Visualization of Geodesic Symmetries (♮ only)

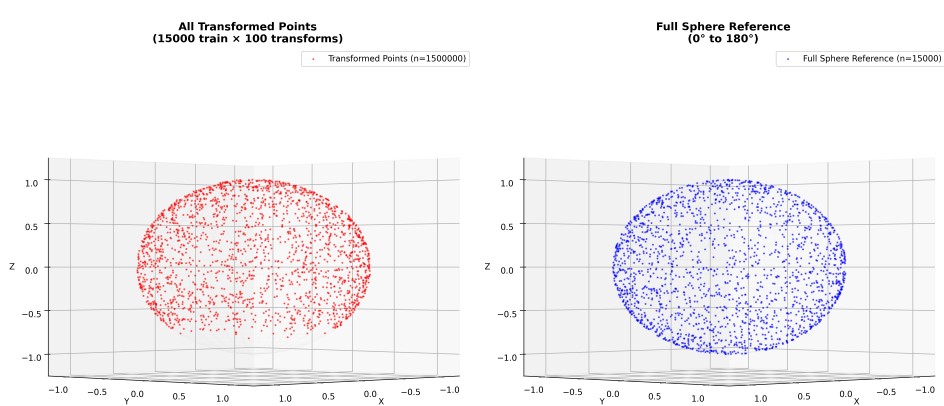

Figure 10: Visualization of Sphere Construction of LieGAN

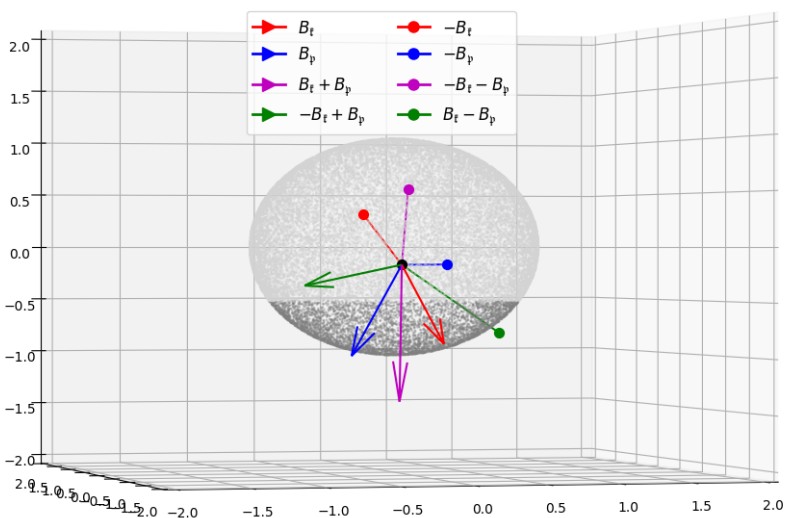

Figure 11: Visualization of Geodesic Symmetry on Tangent Space. Each basis vector (lines ending in arrows) is reflected to its negation (lines ending in circles) by geodesic symmetry.

