# OpenReview forum: "Symmetric Space Learning for Combinatorial Generalization"
_ICLR.cc/2026/Conference — ICLR 2026 Poster_

### Official Review · Reviewer_8EBC · 2025-10-24

**Soundness:** 2
**Presentation:** 2
**Contribution:** 3
**Rating:** 2
**Confidence:** 2

**Summary:**

This paper explores a method for so-called combinatorial generalization. The idea is to assume that the data lies on a manifold $M$ which has the form of a symmetric space, i.e. has the structure $G/K$, where $K \subseteq G$ are Lie groups. $G$ and $K$ are thereby unknown. The authors utilize that such a structure induces a decomposition of the Lie algebra $\mathfrak{g}$ of $G$ into spaces $\mathfrak{l}$ and $\mathfrak{p}$ corresponding to the Lie algebra of the subgroup $K$ and the tangent space of the manifold $M$. Flipping a vector $P\in \mathfrak{p}$ thereby corresponds to changing direction of a geodesic through a chosen origin on $M$.

The authors use facts about the structure of $\mathfrak{l}$ and $\mathfrak{p}$ ($[\mathfrak{l},\mathfrak{l}]\subseteq \mathfrak{l}, [\mathfrak{p},\mathfrak{l}]\subseteq \mathfrak{p}, [\mathfrak{p},\mathfrak{p}]\subseteq \mathfrak{l}]$), and the fact that flipping the $P$-vector corresponds to changing time-direction in geodesics to design loss functions for a learning framework. This is subsequently incorporated into a flow-matching model to produce state-of-the art results on several datasets.

**Strengths:**

The framework the authors propose is grounded in mathematical theory and is elegant.The models have been tested on multiple dataset, and fare well on all of them.

**Weaknesses:**

Let me start by saying that I have little experience on the problem of combinatorial generalization, so I am very open to the authors correcting me/ critiquing my assessment.

With this said, I think that the main weakness of this paper in that it suffers in clarity and reproducibility. This is the main reason for my rating.

1. The 3D sphere shape manifold reconstruction data is not properly described. It is not describe how the training and data set are generated. In fact, it is to me not clear whether the point cloud in (a) is the entire dataset or if (a) is a point in the training dataset -- same for (b). Given the description of the dataset in the appendix, I am inclined to believe that (a) is the entire data set, and that each point in the cloud is a training example, but how is then a PointNet used to process the data point?

It is not clear what the generation process of the points in (a) is, and how the ground truth (b) is fed into the models. If I understand e.g. the dSprites datasets correctly, the net is given a set of latent parameters it has not seen before (shape,scale, orientation, position-x,position-y) and its task is to generate an image corresponding to the latent parameters. What are the latent parameters here?

It should also be noted that the authors have provided no code.

2. I do not understand entirely understand to which extent the GSC-loss is encouraging the decoder to be geodesic-symmetric. In fact, as formulated in 3.2, the loss 'only' encourages the encoder E and decoder D to satisfy $E\circ D\approx \mathrm{id}$ on the flipped versions of the latent vectors of the datasets. Can the authors provide any experimential insight to whether this is actually what happens in practice? This is particularly interesting given that the ablation studies point to the GSC-loss being important.

 In my opinion, providing some type of experimental evidence here is crucial. If the GSC-loss is not actually encouraging the decoder to be geodesic-symmetric, the theoretical considerations of the paper are not providing and explaination to the increased performance of the model.

**Questions:**

See weaknesses.

Also, the symmetry techniques do not bring any increase in performance on the R2R task on the 3D Shapes dataset. Can the authors comment on why they think this is?

---

> ### Author Response · Authors · 2025-11-23
>
> ## Response for Weakness
>
> A1:
> - Fig. 3 (a) is training data, and (b) is test data. This 3D Sphere dataset  is generated by sampling points from a sphere defined by $0\leq\theta\leq\pi, 0\leq\phi\leq2\pi$. The training set consists of the partial arc where $0\leq\theta\leq135^\circ$(Figure 3a), while the test set covers the unseen region $135^\circ<\theta\leq\pi$ (Figure 3b).
> - Regarding the architecture, while PointNet is typically used for processing entire point clouds, in our manifold learning context, we treat each point $x\in\mathbb{R}^3$ as an independent sample from the data distribution. Our encoder processes a batch of individual points, mapping each point to its latent representation $z$, rather than encoding the whole set as a single object. The “PointNet” here refers to the use of shared MLPs to process each point independently.
> - Unlike dSprites, this dataset does not provide explicit latent labels (factors). Instead, the model receives only the cordiantes $(x,y,z)$ and must autonomously learn the intrinsic manifold structure (i.e., spherical coordinates). This experiment explicitly tests whether the model can leverage the $SO(3)$ symmetry structure learned from the partial observation to reconstruct the unseen submanifold. We have revised the paper to clarify this and included the experiment script in the supplementary material.
>
> A2:
> As you correctly pointed out, the GSC loss mathematically enforces $E(D(-p)) \approx -p$. However, this constraint fundamentally relies on the decoder $D$. If $D(-p)$ generates a sample off the manifold or in an incorrect geometric location, the encoder $E$—which is trained to map data to the latent space—would fail to map it back to $-p$. Therefore, to minimize this loss, the decoder is forced to generate valid samples in the unseen region that align with the symmetric structure of the latent space.
> To verify this experimentally, we visualized the decoding results of $-p$ (the antipodal point in latent space). As shown in Figure 8 of the rebuttal PDF, the model trained with GSC correctly generates points on the opposite side of the sphere (the unseen region), whereas the model without GSC fails to do so. This confirms that GSC effectively aligns the geometry of the decoded space with the latent symmetric structure.
>
> ## Response for Questions
> A1:
> - For main table (Table 2.) , our method shows the best performance for every case.
> - We assume you are referring to Cases 2 and 3 in the R2R setting for 3D Shapes in ablation study (Table 3.). You are correct that the performance gain is less pronounced in these specific cases. This is largely due to the complexity of the R2R task on 3D Shapes, which involves intricate combinations of factors, making the model sensitive to hyperparameters.
> - For fair comparison and consistency, we used a **fixed set of hyperparameters** (e.g., $\lambda=1$) across all datasets and settings, rather than fine-tuning for each specific case. Despite this constraint, it is important to note that our method **outperforms baselines in 9 out of 12 scenarios and competitive in 2 scenarios**, demonstrating strong and consistent generalization capabilities overall. The results in the 3D Shapes R2R task suggest that while our method is robust, there is room for further performance maximization through task-specific hyperparameter tuning.

---

> > ### Comment · Reviewer_8EBC · 2025-11-25
> > **Response**
> >
> > 1. Thank you for explaining the setting of your experiment better. In particular what you meant by using a pointnet really threw me off. Would it not be more correct to say that the encoder is an MLP?
> > Am I correct that what you are doing is to check whether $D\circ E(x) = x$ also for $x$ in the bottom part of the sphere (the 'test set')? If this is the case, the experiment convinces me a lot more. I agree that this shows that the encoder really seems to learn the geometry of the underlying manifold. I will therefore raise my score.
> > I reread the parts of the manuscript related to this experiment, and do not think that it describes properly that this is what is done here. A sentence or two describing this would not hurt.
> >
> > 2. The plot in the updated manuscript (which needs to have an explaination and be referred to if it should be kept there!) does not convince me that the GSC-loss actually enforces geodesic symmetry. Do I understand correctly that  the purple dots are the outputs D(-E(x)), where x is in the training set? If the theory drawn up by the authors in the theoretical part really describes what is going on in their model, should this not be points 'on the other side' of the geodesics going from an origin on the data-manifold to x? What happens if you only flip the part of E(x) that is in $\mathfrak{p}$?
> > I am thus not entirely convinceb by the explaination of the authors model, and will therefore not raise my score more for now.
> >
> > 3. I thank you for the responses to my questions, they are satisfactory.
> >
> > 4. A comment to the updated pdf: It seems like you are presenting the results of the LieGan in a completely different manner compared to your model, in particular not reporting the same metric. Is this so, and what is then the reason?

---

> > > ### Author Response · Authors · 2025-12-03
> > >
> > > 1. **Encoder Architecture & Plot Intent**
> > > You are correct that the encoder effectively functions as an MLP. We will revise the manuscript to reflect this as suggested. Furthermore, the provided plot illustrates the distribution of training samples (red points) and their geodesically flipped counterparts (magenta points). Our intention was to demonstrate that valid reconstruction ($D \circ E(x) \approx x$) is achievable even in the unseen region, effectively visualizing the impact of the GSC loss. As you correctly interpreted, this serves to show that both the encoder and decoder have successfully learned the underlying structure of the manifold. We will add this explanation to the manuscript.
> > >
> > > 2. **GSC Loss and Geodesic Symmetry Verification**
> > > You are correct that the magenta points represent the outputs $D(-E(x))$. To empirically demonstrate that the GSC loss effectively induces geodesic symmetry, we have added a new plot in the Appendix. This visualization displays the directions of the learned Lie algebra bases originating from the center, alongside their negated counterparts (showing the best-case alignment).
> > > In the figure, the lines ending in circles (representing the negated basis directions) are positioned almost diametrically opposite to the lines ending in arrows (representing the original basis directions). This alignment is consistent with the Lie algebra-theoretic definition of geodesic symmetry, implying that points on the generated geodesics are effectively reflected through the origin to their antipodal positions.We acknowledge that achieving perfect geodesic symmetry is challenging due to factors such as (1) the symmetry being induced via a soft loss constraint rather than an architectural hard constraint, and (2) potential imbalances in gradient pressure caused by variations in data distribution across different directions relative to the origin. However, the results clearly demonstrate a strong overall tendency toward the desired symmetry.
> > > Furthermore, the observation that the learned basis directions extend not only toward the observed training manifold but also project into the test regions allows us to infer that the symmetry has been successfully propagated to the unseen domain. We will incorporate these analyses and visualizations into the final manuscript.
> > >
> > > 3. **Acknowledgement** We thank you for your understanding and satisfactory response to our previous clarifications.
> > >
> > > 4. **LieGAN Evaluation Metrics** Regarding the LieGAN results, LieGAN is fundamentally a GAN model without conditioning mechanisms, making standard reconstruction tasks impossible in a naive manner. Therefore, we evaluated the results qualitatively by verifying the coverage obtained by applying transformations sampled from the learned symmetry encoder. While direct quantitative comparison using the exact same reconstruction metric is not feasible, evaluating this coverage against the entire ground truth sphere yielded a Chamfer Distance of 0.0072. (Please refer to Table 1; this performance is inferior to our method). We will ensure these details are included in the final manuscript.

---

### Official Review · Reviewer_iY5v · 2025-10-31

**Soundness:** 2
**Presentation:** 3
**Contribution:** 3
**Rating:** 4
**Confidence:** 3

**Summary:**

In this paper, the authors propose a method for addressing the issue of symmetry generalization. The method operates under the assumption of symmetric spaces, using Cartan decomposition and learning a Lie algebra. The geodesic symmetry property of symmetric spaces is used to extend/extrapolate symmetries to unobserved samples in the complete data space--that is, the space in which the observed data lives. The authors also conduct experiments in which their method appears to outperform baselines.

**Strengths:**

1. This paper is built upon a strong mathematical foundation.

2. The experimental results demonstrate the advantages of the proposed methodology compared with the chosen baselines.

3. This paper demonstrates potential for acceptance, provided that the perceived weaknesses can be addressed.

**Weaknesses:**

1. *The issue of symmetry generalization is not properly motivated.* The authors cite a single paper from which they draw the conclusion that current symmetry-based machine learning methods are limited in that learned symmetries do not extend beyond observed samples. In light of previous work that is not considered in this paper, the statement on the symmetry generalization challenge is actually not true. The authors are encouraged to discuss their statement in light of additional recent work on symmetry discovery, particularly for the following sources: "Generative Adversarial Symmetry Discovery" (Yang et al., 2023), "Learning Infinitesimal Generators of Continuous Symmetries from Data" (Ko et al., 2024), and "Symmetry Discovery Beyond Affine Transformations" (Shaw et al., 2024). On the other hand, the authors may be able to justify the exclusion of certain sources. But as written, the symmetry generalization statement does not appear to be reflective of recent work.

2. *Additional methods to compare with.* From an experimental standpoint, it seems there are other geometry-inspired autoencoders that should be compared with. One such method is "Geometry Regularized Autoencoders" (Duque et al., 2023), and another is "Geometry-aware generative autoencoders for warped riemannian metric learning and generative modeling on data manifolds" (Sun et al., 2024).

**Questions:**

1. It seems that the perceived issue of symmetry generalization is being solved primarily by making restrictive assumptions about the structure of the symmetry group. Generally, restrictive assumptions can make problems more tractable, but I am concerned that the limitation to transitive group actions is too restrictive. After all, the action of rotations in the plane about the origin is not a transitive group action: this seems like one of the simplest possible symmetries to consider. Can the authors speak to this?

2. The notion that there are points in the complete space that are not mapped to via the observed group acting on the observed data seems to be a model assumption, which assumption seems equally valid to the assumption that training data defines the manifold and/or distribution of data. It seems that the notion that test data is not only out of distribution, but also that it aligns with the symmetric space model, is a model assumption that is not motivated by information obtainable from the training data: in fact, methods which rely on the training data are prone to inherit the generalization challenge spoken of. Are these correct statements? If so, can the authors speak to (or reiterate) justification for these model assumptions?

---

> ### Author Response · Authors · 2025-11-22
> **Response for Weakness**
>
> A1:
> - We appreciate you highlighting these relevant works. We agree that Symmetry Discovery is a vital field and will include the suggested references (Yang et al., Ko et al., Shaw et al.) in our revised related work.
> - However, we respectfully disagree that the challenge of "Symmetry Generalization" is unfounded. Existing symmetry discovery methods such mentioned above typically aim to find the group structure that best **describes the support of the observed training data**. While effective for fully observed manifolds (like MNIST digits), these methods face limitations under **partial observation**. As demonstrated in our experiment, descriptive methods tend to learn only the local symmetries valid within the observed region, without a mechanism to extrapolate to the global symmetry group.
> - In contrast, our goal is **combinatorial generalization (CG)**, which requires inferring unseen combinations. We tackle this by imposing a strong **geometric inductive bias (Symmetric Space)**. This allows us to not just describe the observed data, but to actively **construct and enforce** symmetries in the unseen regions via Cartan decomposition and geodesic symmetry. Thus, our work complements existing discovery methods by focusing on **extrapolation** rather than description.
>
> A2:
> - We appreciate your suggestion to compare our method with additional geometry-inspired baselines, including "Geometry Regularized Autoencoders" (GRAE, Duque et al., 2023) and "Geometry-aware generative autoencoders" (Sun et al., 2024).
> - We confirm that we have revised **Table 2** to include experimental results for the architecture presented in Sun et al. (2024).
> - Regarding the Geometry Regularized Autoencoders (GRAE), we encountered significant technical difficulty and time consumption in porting the method and thus could not complete the experiments for the encouraged responding deadline. We aim to include the GRAE results in a subsequent revision, but based on the strong performance against other symmetry-aware models already included, we believe our current comparative analysis is sufficiently validated.

---

> > ### Comment · Reviewer_iY5v · 2025-11-24
> >
> > I thank the authors for their comprehensive response to the initial review. So far, they have addressed the majority of my questions and concerns. There is still a lingering concern I have, which concern may possibly be alleviated by further discussion.
> >
> > I believe I understand the mechanism by which this paper obtains symmetries which generalize to unobserved data. But existing methods may obtain generalization through different and more simple (or so apparently) restrictive assumptions. For example, the first recommended reference was that of LieGAN: in this paper, the authors performed an experiment in which rotational symmetry was discovered from partial trajectories. It seems that large sections of circles were unobserved--and yet, the authors claim to have discovered a symmetry from the observed data alone. It seems to me that generalization can occur, as in the present paper, under restrictive assumptions. In the case of LieGAN, this assumption is that the infinitesimal generator(s) which corresponds to the symmetry are linear combinations of a pre-determined Lie algebra basis. Similarly for the second reference, the assumption is that the infinitesimal generators are linear combinations of a collection of pre-determined vector fields.
> >
> > The restrictive assumptions made at present appear to be new, and there may not be an exact overlap of the problems methods such as LieGAN can generalize to and problems which the current method may generalize to. But I am still not convinced that the symmetry generalization problem is as persistent as the authors seem to suggest. To test this explicitly, I propose that the authors test LieGAN (Yang et al.) and the two-step method in Shaw et al. using the the 3D sphere data. In the case of LieGAN, my hypothesis is that the training data will yield a description of symmetry which is also applicable for the test data, owing to the restrictive assumption that the infinitesimal generators are linear combinations of the infinitesimal generators of SO(3). And for Shaw et al., my hypothesis is that the first step (so-called "level set estimation") will yield a manifold characterization similar to $x^2+y^2+z^2-1=0$, owing to the restrictive assumption that the component(s) of the level set are quadratic polynomials, which characterizes the manifold on which not only the training set resides, but also the test set. Subsequently, the vector field infinitesimal generators computed in the second step would describe symmetry not only of the training set, but also of the test set, owing to the assumption that such a vector field is a linear combination of the vector fields corresponding to affine symmetries in $\mathbb{R}^3$.
> >
> > If I am understanding the issue of symmetry generalization correctly, my interpretation of the authors' contribution is that they are introducing a new restrictive assumption, which is that the underlying manifold is assumed to be a symmetric space. I think this is an interesting and novel idea, but my opinion is that the statement that current methods cannot generalize to unobserved data is not correct, since other methods appear to generalize to unobserved regions by making parametric assumptions about the form of the infinitesimal generators.

---

> ### Author Response · Authors · 2025-11-22
> **Response for Qeustions**
>
> A1:
> - You raise a valid point regarding rotation in $\mathbb{R}^2$ fixing the origin. However, our framework models the **latent data manifold itself**, not the ambient space. For example, if the data lies on a circle $S^1$ (embedded in $\mathbb{R}^2$), the action of the rotation group $SO(2)$ on the manifold $S^1$ is indeed **transitive**.
> - In the context of Combinatorial Generalization, the latent space is typically modeled as a product of factors (e.g., shape $\times$ color $\times$ orientation). The symmetry group acting on this space can be viewed as a direct product of transformation groups for each factor. Since each factor's variation (e.g., shifting position, changing hue) covers its respective submanifold transitively, the assumption of a transitive group action on the latent manifold is both natural and necessary to navigate from seen to unseen combinations.
>
> A2:
> - We confirm that your statements regarding our model assumptions are **correct**. We explicitly make these assumptions to address the fundamental challenge of generalization. Below, we provide our justification for why these assumptions are both necessary and valid in the context of Combinatorial Generalization (CG).
>     1. Necessity of Inductive Bias for Generalization
>     We agree that methods relying solely on information obtainable from training data are inherently prone to failure in generalization tasks, as the training support strictly contains no information about the unseen regions. To bridge this gap, it is essential to inject a strong inductive bias.
>     - **Assumption 1 (Existence of Unseen Mapped Points):** We assume the complete data space contains points unreachable by the *observed* group action alone. This is the definition of the CG problem; standard models fail because they implicitly assume the local group structure learned on $X_{obs}$ is sufficient, whereas we posit that a global structure must be extrapolated.
>     2. **The Manifold Hypothesis and Symmetric Space Prior**
>     - **Assumption 2 (Alignment with Symmetric Space):** You are correct that assuming the test data aligns with our learned symmetric space is a strong assumption. Standard Empirical Risk Minimization (ERM) does not guarantee this alignment.
>     - **Justification:** Our justification is rooted in the **Manifold Hypothesis**. We assume that both observed (training) and unobserved (test) data lie on the same underlying low-dimensional manifold, governed by consistent semantic or physical laws (symmetries).
>         - Instead of hoping the model "unintentionally" aligns with the unseen data, our **CartanFM** framework **explicitly enforces** this alignment. By modeling the manifold as a **Symmetric Space**, we provide the minimal but rigorous geometric structure required to extrapolate symmetries consistently from the observed region to the unobserved region (via operations like geodesic symmetry).
>         - Our experiments empirically validate this assumption: the model successfully captures the global structure (e.g., the full sphere) from partial observations, demonstrating that the "Symmetric Space" assumption is a highly effective inductive bias for CG tasks.

---

> ### Author Response · Authors · 2025-11-25
>
> We appreciate your insightful comment and the proposal to empirically test existing methods. We prioritized the experiment with LieGAN (Yang et al.) as suggested. Unfortunately, we could not evaluate the method by Shaw et al. within this timeframe due to the absence of an official code implementation.
> Experimental Details:
> - Dataset: The model was trained on the same partial sphere dataset, consisting of 15,000 points sampled from the upper region ($0 \le \theta \le 135^\circ$).
> - Initialization: The Lie algebra basis of the model was randomly initialized to simulate a true "discovery" setting.
> - Architecture: The discriminator utilized a PointNet backbone followed by linear layers.
>
> Results and Analysis:The visualized result Fig.9 is included in the revised Appendix D. Given the difference between our VAE-based approach and the GAN framework, we evaluated generalization by applying 100 sampled transformations to the data and inspecting whether these transformations could populate the unseen test region ($135^\circ < \theta \le \pi$).
>
> Our observations indicate that LieGAN failed to generate the test region effectively. Furthermore, contrary to the hypothesis that the model would recover the correct group structure, the learned basis matrices did not converge to a skew-symmetric form (please refer to the matrix below).
>
> $$
> \begin{bmatrix}
>  0.00263261 & -0.0025511 &  -0.03792508\\\\
>  0.00370286 &  0.000771 &   -0.07683899\\\\
>  0.03461159 &  0.06822063 &  0.00352986\\\\
> \end{bmatrix}
> $$
>
> Conclusion:
> This result implies that the parametric assumptions of existing discovery methods (e.g., linear combinations of bases) are insufficient to guarantee extrapolation in partial observation settings. Without a stronger geometric inductive bias—like our Symmetric Space assumption—the model struggles to identify the true global symmetry ($SO(3)$) among many possible local solutions that fit the partial data. We believe this empirically validates the necessity and contribution of our proposed method.

---

> > ### Comment · Reviewer_iY5v · 2025-11-26
> >
> > I appreciate the additional experimental result the authors have conducted. In my opinion, this increases the standing of the paper, and I have no remaining reasons to object to the acceptance of this paper and will adjust the score accordingly.
> >
> > It is unfortunate that an official implementation of the other method is missing, discouraging widespread use (on the other hand, the authors claim that the method may be conducted merely using polynomial regression with constrained coefficients). However, I feel that the use of LieGAN is sufficient to motivate the problem of symmetry generalization.

---

### Official Review · Reviewer_WXcp · 2025-11-10

**Soundness:** 3
**Presentation:** 3
**Contribution:** 3
**Rating:** 6
**Confidence:** 3

**Summary:**

This paper addresses a key challenge in symmetry learning—how to generalize from training data to unseen data. It introduces the Cartan loss and the Geodesic Symmetry Consistency loss, which are designed to enforce fundamental properties of Lie algebras and enhance generalization capabilities, respectively. The learnable components include a Lie Algebra Encoder, which maps input data to coefficients of Lie algebra bases; learnable Lie algebra bases; and a conditional flow matching model, which generates a vector field based on the Lie algebra and data points. Across a series of combinatorial generalization benchmarks, the proposed method consistently outperforms existing baselines.

Given the theoretical rigor and innovation, I am inclined to accept it.

**Strengths:**

- To my knowledge, none of the existing symmetry discovery methods take into account the fundamental property of Lie algebras—the relationship defined by the Lie bracket. The proposed CartanFM in this paper rigorously incorporates this aspect, resulting in a more reasonable learned Lie algebra space, which I consider one of the highlights of this work.

- Most symmetry discovery approaches opt to directly learn either the basis of the Lie algebra or the vector field. The former is often limited to relatively simple linear symmetries, while the latter, due to the complexity of vector fields, generally lacks interpretability. CartanFM’s flow matching module innovatively addresses both limitations. Specifically, it establishes a mapping between the Lie algebra and the vector field, enabling us to explicitly solve for the Lie algebra basis while also obtaining its specific action form on data points.

- The generalization capability on unseen data provides potential for scientific discovery and the learning of structured representation spaces.

**Weaknesses:**

- Due to the involvement of learning Lie algebra bases, a brief discussion and comparison of the related symmetry discovery work (https://arxiv.org/abs/2310.00105, https://arxiv.org/abs/2410.21853, https://arxiv.org/abs/2403.01946, https://arxiv.org/abs/2510.01855) is necessary.

- It is better to list the contributions point by point in detail in Section 1.

**Questions:**

- I notice that Definition 2.2 is defined for a one-parameter group action ($t \in \mathbb{R}$). So how does it guide the design of the GSC loss in the multi-parameter case?

- Does this method have potential for application in high-dimensional data—such as in the embedding spaces of images and text?

---

> ### Author Response · Authors · 2025-11-22
>
> ## Response for Weakness
> A1:
> - Thank you for this point. We agree that a comparative discussion with recent symmetry discovery work is necessary due to our shared foundation in Lie algebra bases. We have revised the **Related Work** section to incorporate the suggested references and clarify the distinction in methodology and goals.
>     - **Fundamental Difference:** We clarify that the **Symmetry Discovery** task is inherently **descriptive**, aiming to find the group structure that best explains the support of the *given data*. Consequently, when faced with **partial observation**, these methods lack the explicit mechanism to guarantee extension into the unseen regions.
>     - **Our Goal:** In contrast, our goal is **extrapolative**. Our method introduces the geometric **Symmetric Space prior** to explicitly force the extension of the learned symmetry structure beyond the boundaries of the observed data manifold.
>
> A2:
> We acknowledge the suggestion to improve the structure of the introduction. We have revised Section 1 to clearly list our contributions point-by-point, enhancing the clarity and impact of our work.
>
> ## Response for Questions
> A1:
> - We appreciate the reviewer's keen eye on the theoretical details.
> - Definition 2.2 defines geodesic symmetry $s_p$ using a geodesic curve $\gamma(t)$, which is indeed a one-parameter curve. However, a key property of Riemannian symmetric spaces is that this local symmetry $s_p$ induces a linear map on the tangent space $T_p\mathcal{M}$ (which we identify with $\mathfrak{p}$), specifically $v \mapsto -v$ for **any** tangent vector $v \in T_p\mathcal{M}$.
> - Since any tangent vector $P \in \mathfrak{p}$ generates a unique geodesic $\gamma(t) = \exp_p(tP)$, the reflection principle $\gamma(t) \to \gamma(-t)$ corresponds directly to the negation of the tangent vector $P \to -P$ in the Lie algebra. This relationship holds regardless of the dimensionality of the tangent space $\mathfrak{p}$.
> - Therefore, in the multi-parameter case (where $P$ is a high-dimensional vector), the GSC loss generalizes this principle by enforcing consistency between the generated sample from the negated vector $-P$ and the negation of the original vector $P$. Essentially, the negation operation $-P$ is the algebraic realization of geodesic symmetry for the entire multi-dimensional vector field.
>
> A2:
> - This is an insightful question regarding the broader applicability of our method.
> - Yes, we believe our framework has strong potential for high-dimensional data such as image or text embeddings. Our method operates on the **latent manifold** learned by an encoder, not directly on the raw high-dimensional input space. According to the Manifold Hypothesis, even high-dimensional data (like images or text) lie on a lower-dimensional intrinsic manifold.
> - As long as we can assume that this latent manifold possesses a symmetric structure (or can be approximated by one), our method can be applied to learn the underlying Lie algebra and extend symmetries. For example, applying CartanFM to the latent space of large-scale pre-trained models (e.g., CLIP embeddings or Diffusion latents) could be a promising direction to discover and generalize semantic transformations (e.g., style transfer or arithmetic operations on concepts) in a principled geometric way.

---

> > ### Comment · Reviewer_WXcp · 2025-11-24
> >
> > Thank you for your reply. My concerns are addressed, and I am happy to raise my score. By the way, I suggest using LaTeX's "change" package to highlight the revisions when submitting the revised manuscript next time.

---

### Meta-Review · Area_Chair_J9KR · 2026-01-07

**Summary:**

1. The need for symmetry generalization is not well motivated (**iY5v**)
2. Missing baselines (**iY5v**)
3. Transitive group assumption is too limiting.  The model assumptions need to be clarified and related to the assumptions of other symmetry discovery methods. (**iY5v**)
4. The method is not sufficiently clearly described for reproducibility. (**8EBC**)
5. Provide evidence the GSC-loss encourages the decoder to be geodesic-symmetric (**8EBC**)

**Reviewer Concerns:**

1. Additional justification was added.
2. One of two suggested baselines were added.
3. The authors further justified their assumptions and ran a comparison with LieGAN in a partially observed symmetry setting.
4. The authors clarified their method and discussed changes to the explanation with the reviewer.
5. The authors added a new experiment showing geodesic-symmetry is successfully induced.

**Reviewer Scores:**

- *WXcp* increased their score from 6 to 8 after the authors answered their questions, edited the paper, and  better cited the literature.
- **iY5v** raised their score from 4 to 8.
- **8EBC** gave a 2, asked many questions, and signaled they were open to discussion.  The authors addressed some of their questions and they increased their score to 4.  They asked more questions which the authors also answered.  They may have increased their score to 6.

---

### Decision · Program_Chairs · 2026-01-26

Accept (Poster)